# Effect of Polycyclic Aromatic Hydrocarbons on Development of the Ascidian *Ciona intestinalis* Type A

**DOI:** 10.3390/ijerph17041340

**Published:** 2020-02-19

**Authors:** Toshio Sekiguchi, Hiroshi Akitaya, Satoshi Nakayama, Takashi Yazawa, Michio Ogasawara, Nobuo Suzuki, Kazuichi Hayakawa, Shuichi Wada

**Affiliations:** 1Noto Marine Laboratory, Institute of Nature and Environmental Technology, Division of Marine Environmental Studies, Kanazawa University, Noto-cho, Ishikawa 927-0553, Japan; t-sekiguchi@se.kanazawa-u.ac.jp (T.S.); nobuos@staff.kanazawa-u.ac.jp (N.S.); 2Department of Animal Bioscience, Faculty of Bioscience, Nagahama Institute of Bio-Science and Technology, Nagahama, Shiga 526-0829, Japan; b213001@m.nagahama-i-bio.ac.jp; 3Department of Biology, Graduate School of Science, Chiba University, 1-33 Yayoi-cho, Inage-ku, Chiba 263-8522, Japan; snakayama@chiba-u.jp (S.N.); ogasawara@faculty.chiba-u.jp (M.O.); 4Department of Biochemistry, Asahikawa Medical University, Hokkaido 078-8510, Japan; yazawa@asahikawa-med.ac.jp; 5Low Level Radioactivity Laboratory, Institute of Nature and Environmental Technology, Kanazawa University, Nomi City, Ishikawa 923-1224, Japan; hayakawa@p.kanazawa-u.ac.jp

**Keywords:** polycyclic aromatic hydrocarbons, development, ascidian, *Ciona intestinalis* type A, marine invertebrates

## Abstract

Polycyclic aromatic hydrocarbons (PAHs) are pollutants that exert harmful effects on marine invertebrates; however, the molecular mechanism underlying PAH action remains unclear. We investigated the effect of PAHs on the ascidian *Ciona intestinalis* type A (*Ciona robusta*). First, the influence of PAHs on early *Ciona* development was evaluated. PAHs such as dibenzothiophene, fluorene, and phenanthrene resulted in formation of abnormal larvae. PAH treatment of swimming larva induced malformation in the form of tail regression. Additionally, we observed the *Ciona*
*aryl hydrocarbon receptor* (*Ci-AhR*) mRNA expression in swimming larva, mid body axis rotation, and early juvenile stages. The time correlation between PAH action and *AhR* mRNA expression suggested that Ci-AhR could be associated with PAH metabolism. Lastly, we analyzed *Ci-AhR* mRNA localization in *Ciona* juveniles. *Ci-AhR* mRNA was localized in the digestive tract, dorsal tubercle, ganglion, and papillae of the branchial sac, suggesting that Ci-AhR is a candidate for an environmental pollutant sensor and performs a neural function. Our results provide basic knowledge on the biological function of Ci-AhR and PAH activity in marine invertebrates.

## 1. Introduction

Polycyclic aromatic hydrocarbons (PAHs) are chemicals that contain multiple aromatic rings and primarily are environmental contaminants [1,2]. PAHs are released into the air via incomplete combustion of wood, coal, and petroleum products [1,2]. Airborne PAHs enter the ocean via both dry and wet precipitation [3]. In addition, since PAHs are present in crude petroleum, accidental oil spillage from oil tankers and offshore oilfield results in severe PAH contamination in the ocean [4,5]. Polluted seawater containing PAHs affects aquatic animals. PAHs were reported to induce oncogenesis, immune system disorders, and developmental defects in aquatic animals [6,7]. Therefore, it is essential to investigate the effect and mechanism of action of PAHs on marine invertebrates to predict their biological impact on the marine environment.

PAHs exert harmful effects on marine invertebrates [8,9,10]. PAHs are associated with DNA damage and oxidative stress in the rock oyster *Saccostrea cucullata* [9]. Phenanthrene and pyrene disturb the immune response in the European clam *Ruditapes decussatus* [8]. Chrysene functions as an endocrine disruptor and immunotoxic substance in the marine crustacean *Penaeus monodon* [10]. PAHs function as oxidative stressors and reduce the gamete number in adult sea urchins [11,12]. Furthermore, PAHs are associated with developmental anomalies [13,14]. Benzo[*a*]pyrene (BaP) induces the exogastrula by ectopic β-catenin action [14]. We previously reported that benzo[*a*]anthracene (BaA), a PAH, suppresses larval spicule formation during early development of the Japanese green sea urchin, *Hemicentrotus pulcherrimus* [15,16]. This larval skeleton malformation is induced by 4-hydroxy BaA (4OH-BaA) as well, which is an internal metabolite in the BaA detoxication pathway. Furthermore, 4OH-BaA was detected in sea urchin embryos treated with BaA [16]. These findings strongly suggested that sea urchins follow the PAH detoxication pathway and generate harmful metabolites such as 4OH-BaA.

The metabolic pathway of PAHs in mammals has been studied extensively with respect to human health [17]. These studies reveal that PAHs exert effects through induction of PAH metabolic intermediates [1,18,19,20]. For example, oncogenesis induced by BaP exposure results from formation of BaP diol epoxides (BPDEs). BaP is recognized by AhR in the cytoplasm. After binding to BaP, AhR is translocated into the cell nucleus and subsequently transactivates monooxygenases genes, including cytochrome P450 family 1 subfamily A member 1 (CYP1A1) [21,22]. The CYP1A1 enzyme and epoxide generate BPDEs from BaP. BPDEs bind to the DNA and form adducts, further inducing DNA mutations that may give rise to tumors [1,20]. Furthermore, the monohydroxylated BaP synthesized from BaP by the CYP1 enzymes exhibits estrogenic or anti-estrogenic functions [18,19]. Therefore, investigation of the PAH metabolic pathway is essential to elucidate the biological function of PAHs. In other words, AhR plays a central role in the PAH detoxication pathway. However, the biological function of AhR and PAH detoxication pathway in marine invertebrates remains elusive.

To investigate the issues associated with marine invertebrates, we studied an ascidian, *Ciona intestinalis* type A (*Ciona robusta*) that is a marine invertebrate model organism. Since the genomic information of the *Ciona intestinalis* type A is readily available [23,24], we could perform comprehensive analyses, such as transcriptomic and proteome analyses, on PAH-inducible or repressible genes [25,26]. The biological functions of PAH detoxication-related genes, including *AhR* and *CYP*, can be investigated due to the availability of transgenic and genome-edited ascidians [27,28,29]. Therefore, *Ciona intestinalis* type A is suitable for the investigation of the molecular mechanism of the PAH metabolic pathway.

Here, we first evaluated the effect of PAHs on early development and metamorphosis of *Ciona intestinalis* type A. We further analyzed the *Ciona* ortholog of the AhR gene as a candidate for a PAH sensor. The expression of *Ciona*
*AhR* (*Ci-AhR*) mRNA was observed in swimming larva, metamorphotic individuals, and juveniles.

## 2. Materials and Methods 

### 2.1. *Animals*

*Ciona intestinalis* type A was purchased from the National BioResource Project and maintained in artificial seawater at 20 °C under constant light to avoid spontaneous spawning. Eggs and sperm were surgically collected from the adult gonoduct. Insemination was performed in artificial seawater. Embryos were reared in the dark at 18 °C for the exposure experiment. Under these conditions, the first cleavage in embryos occurred 1 h after fertilization and the embryos transformed into swimming larvae 18 h after fertilization. Swimming larvae metamorphose approximately 24 h after hatching. According to the developmental stage guideline defined by “Four-Dimensional Ascidian Body Atlas 2” (https://www.bpni.bio.keio.ac.jp/chordate/faba2/ja/top.html) [30], RT-PCR samples from swimming larvae, mid body axis rotation, and early juvenile stages were collected and dissolved in TRIZOL reagent (Thermo Fisher Scientific, Waltham, MA, USA) at 20, 42, and 66 h after fertilization, respectively. All specimens were stored at −80 °C until use. To obtain the whole-mount *in situ* hybridization (WISH) sample, juvenile specimens 1–2 mm in length were collected and fixed in buffer containing 4% paraformaldehyde at 4 °C, as described by Ogasawara et al. [31]. After dehydration using ethanol, all specimens were maintained at −20 °C in 80% ethanol until the experiment.

### 2.2. Chemicals

Dibenzothiophene, fluorene, and phenanthrene were purchased from Sigma-Aldrich (St. Louis, MI, USA). Each reagent was dissolved in dimethyl sulfoxide (DMSO) at a concentration of 50 mM to obtain stock solutions. 

### 2.3. Exposure Experiments

Stock solutions were serially diluted with DMSO to create a dilution series. A total of 0.2% of each solution was added to artificial seawater warmed to 40 °C to produce exposure solutions that were subsequently used at 18 °C. Nominal test concentrations were 100, 50, 25, 12.5, and 6.25 μM. Artificial seawater containing 0.2% DMSO was used as a solvent reference. Exposures were carried out in 6-well plastic tissue culture plates. Each well contained 10 ml exposure solution. Twenty microliters of artificial seawater containing approximately fifty fertilized eggs or swimming larvae were added to the exposure solutions. Fertilized eggs were collected before the first cleavage and used for conducting exposure experiments from fertilized eggs to swimming larvae. Embryos were fixed 22 h after fertilization by adding paraformaldehyde solution to artificial seawater and evaluated for rates of malformation. For exposure experiments on swimming larvae to early juvenile stages, swimming larvae were collected 20 h after fertilization and used in the exposure experiments. During the exposure experiments, larvae metamorphosed into early juveniles. Early juveniles were fixed 90 h after fertilization and examined for the ratio of tail residue length to body length. The exposure experiment was repeated five times, and the average and standard deviation of the measurement results were calculated. The significance test was performed between multiple groups using one-way analysis of variance appropriately. Moreover, the Dunnett’s test was used to determine post-hoc differences. The significance level was accepted at *p*-value < 0.05.

### 2.4. Reverse-Transcription (RT)-PCR of Ci-AhR mRNA

Total RNA was extracted at the swimming larvae, mid body axis rotation, and early juvenile stages using TRIZOL reagent (Thermo Fisher Scientific). cDNA was synthesized from 500 ng total RNA using Superscript III (Thermo Fisher Scientific) and oligo dT(20) primer. *Ci-AhR* cDNA was obtained by RT-PCR using EX-taq polymerase (Takara, Shiga, Japan). Since *Ci-AhR* was previously identified and annotated by Satou and Satoh [32], the primers were designed from the *Ci-AhR* cDNA sequence deposited in GenBank™/EBI Data Bank (Accession number, AB210300.1). *Ci-EF1α* (Accession number, AK113163.1) was used as an internal control for RT-PCR to confirm the quality and quantity of cDNA. *Ci-AhR* primers (forward primer: 5’-CCAGCAAGTCAAACCCAAGC-3’; reverse primer: 5’-AAGTCCCGGTCTTCAGCATG-3’) and *Ci-EF1α* primers (forward primer: 5’-TTTCGCTGTCCGTGACATGA-3’; reverse primer: 5’-CCGCAACCAACCACGTTAAG-3’) were selected using the primer3 Plus web site [33]. The PCR conditions were as follows: 94 °C for 1 min, 35 cycles of 94 °C for 30 s, 60 °C for 30 s, 72 °C for 1 min, and final extension at 72 °C for 7 min. The PCR products were detected using electrophoresis on 1.5% agarose gel. Sanger sequence analysis was performed to confirm that the PCR fragments isolated were *Ci-AhR* and *Ci-EF1α*.

### 2.5. Whole-Mount In Situ Hybridization Analysis of Ci-AhR

cDNA was synthesized from the total RNA extracted from juvenile specimens and used as the PCR reaction template. The *Ci-AhR* PCR fragment (nt 53-3220 of cDNA clone cidg833e11) was amplified using the *Ci-AhR* gene-specific primer set (Ci-AhR forward: 5’-GTTCGTTAGCAGCGGAATTT-3’; Ci-AhR reverse: 5’-CAGCAATGTGTGGGAGAAAA-3’) and subcloned into the pGEM-T easy vector (Promega, Madison, WI, USA). Digoxigenin (DIG)-labeled RNA probes of *Ci-AhR* were synthesized using a DIG RNA labeling kit (Roche Applied Science, Indianapolis, IN, USA). WISH of juvenile *Ciona* specimens was performed as previously described [34,35,36]. 

## 3. Results

### 3.1. Exposure Experiments from Fertilized Eggs to Swimming Larvae

In ascidian development, the embryo first develops into a swimming larva, which then undergoes metamorphosis to form an adult-like juvenile. Swimming larvae consist of a trunk and a tail. The trunk contains the brain vesicle and adult organ primordia, and the tail contains the notochord and muscles. During metamorphosis, the ascidian larvae attach to the substratum via the adhesive papillae on the trunk, absorb the tail, and develop adult organs in the trunk. Here, we examined the effects of PAHs (dibenzothiophene, fluorene, and phenanthrene) on early ascidian development from fertilized eggs to swimming larvae and during metamorphosis from swimming larvae to juveniles.

First, the larvae were treated with PAH, starting at the 1-cell stage to the swimming larva stage (20 h after fertilization). All three examined PAH types exerted a concentration-dependent adverse effect on larval development. While normal larvae have a straight tail, PAHs induce bending in the tail of larvae. Compared to the control, dibenzothiophene and fluorine administered at 100 µM and 50 µM significantly increased the percentage of abnormal larvae (Figure 1a,b). Phenanthrene significantly increased the percentage of abnormal larvae at concentrations of 100, 50, and 25 µM (Figure 1c).

### 3.2. Exposure Experiments from Swimming Larvae to Early Juvenile Stages

PAH treatment was conducted on larvae, starting with swimming larvae 20 h after fertilization to early juveniles 90 h after fertilization, and the effect of PAH on metamorphosis was examined. The effect of PAHs on the rate of normal metamorphosis could not be determined because the metamorphosis rate varied randomly. Next, we focused on the ratio of tail residue length to body length in early juveniles. The tail residue is a mass of tail cells absorbed by the trunk at the beginning of metamorphosis, and it normally shrinks as metamorphosis progresses. The ratio of the tail residue diameter to the juvenile body length was measured, and the measurement results were compared between the early juveniles treated with 100 µM of each PAH and the control early juveniles. The ratio of the diameter of the tail residues to the body length was significantly higher in the PAH-treated early juveniles than in the control early juveniles (Figure 2). In particular, dibenzothiophene-treated early juveniles exhibited the greatest difference compared to control early juveniles. The results described above suggest that the three types of PAHs adversely affect tail absorption during metamorphosis.

### 3.3. Gene Expression of Ci-AhR during Early Metamorphosis

In vertebrates, AhR is a sensor protein of toxic agents such as PAHs and 2,3,7,8-tetrachlorodibenzodioxin [21,22]. After AhR binds PAHs, AhR induces *CYP1A1* gene transcription, which is a monooxygenase involved in PAH detoxication [21,22]. Since Ci-AhR is likely to be a putative PAH sensor as well, we evaluated *Ci-AhR* gene expression during metamorphosis. RT-PCR analysis revealed that *Ci-AhR* mRNA expression was detected in the swimming larval, mid body axis rotation, and early juvenile stages (Figure 3).

### 3.4. Localization Analysis of Ci-AhR mRNA Using WISH

To further evaluate the expression pattern of *Ci-AhR*, we performed WISH analysis in *Ciona* juveniles. A strong *Ci-AhR* mRNA signal was observed in the digestive tract (Figure 4a). *Ci-AhR* mRNA was expressed in the anterior part of the esophagus (Figure 4c). The *Ci-AhR* transcript was detected in the entire stomach (Figure 4d). *Ci-AhR* mRNA was localized in the middle and posterior part of the intestine (Figure 4e,f). Moreover, the *Ci-AhR* gene was transcribed in the ganglion and dorsal tubercle (Figure 4g). *Ci-AhR* mRNA expression was detected in the gill papillae (Figure 4h). Lack of signal detection in the experiment using sense probe indicates that signals developed using antisense probe are specific (Figure 4b).

## 4. Discussion

Marine invertebrates inhabit various marine environments, such as the intertidal zone, shallow sea, and deep-sea regions, and they are consumed as aquatic resources. The influence of marine pollutants on marine invertebrates is a vital subject in research on environmental preservation and marine resource conservation. PAH is a pollutant that is used in the investigation of biological effects owing to its carcinogenic, mutagenic, endocrine disruptive, and immunotoxic properties. However, the biological action and metabolic pathway of PAHs in marine invertebrates is yet to be characterized completely. Therefore, we have focused on these issues using *Ciona intestinalis* type A as a model animal for marine invertebrates.

We first evaluated the effect of PAHs on early development. Treatment of the fertilized egg with dibenzothiophene, fluorene, and phenanthrene resulted in the formation of abnormal larvae in a dose-dependent manner (Figure 1). Bellas et al. [13] examined the effects of PAHs on invertebrates, including *Ciona intestinalis*. They treated *Ciona* embryos with phenanthrene at stages ranging from fertilized eggs to swimming larvae and did not observe any toxic effects. Since the concentrations of phenanthrene they used were lower than the concentrations we used, their results are consistent with our results. Considered together, these results revealed that PAHs affect early *C. intestinalis* type A development. Furthermore, this study is the first to denote the toxic effect of dibenzothiophene on early development. Next, the effect of PAHs on metamorphosis was elucidated. Consistent with the results for early development, an abnormality was observed in early juveniles treated with dibenzothiophene, fluorene, and phenanthrene. Even though the metamorphosis defect was weaker than that in early development, there was an increase in the ratio of cell debris formed by tail regression to body length (Figure 2). *Ciona* larvae swim for several hours to several days before they attach to a substrate, following which metamorphosis begins. Larval tail regression and body axis rotation occur successively [37]. The larval body plan converts to a juvenile body plan [37]. Debris derived from the larval tail is absorbed during the early juvenile stage [37]. The PAH-induced phenotype is expected to be induced by the suppression of larval tail absorption. Overall, we unveiled that PAHs disturb *Ciona* metamorphosis. Collectively, dibenzothiophene, fluorene, and phenanthrene exerted a toxic effect on *Ciona* development.

The metabolic pathway of PAHs has been characterized in vertebrates. Vertebrate AhRs bind to PAHs in the cytoplasm, following which AhR activates the transcription of the *CYP1* family gene. PAHs are metabolized by the CYP1 family protein and eventually discharged from cells. Conversely, the PAHs metabolic pathway via AhR in marine invertebrates remains elusive and hence is a subject of significant interest. Recent genomic research has revealed that the *AhR* gene broadly exists in eumetazoa, including bilaterians, cnidarians, and placozoans [38]. Further, the expression of both *AhR* and *CYP* genes is induced by PAH treatment of marine invertebrates [39,40]. Since there are no reports of PAH binding to AhR [38], it is unclear how PAHs activate AhR in marine vertebrates. Induction of *AhR* and *CYP* genes by PAHs suggests that PAHs activate AhR by mechanisms yet to be determined, and AhR controls expression level of *CYP* genes as a transcription factor.

We selected *Ciona* as a model animal to investigate the biological and molecular functions of AhR in marine invertebrates as there are experiment systems to investigate the molecular function of transcriptional factors like Ci-AhR [41]. Gain-of-function and loss-of-function experiments using overexpression and genome-editing systems are available, respectively [27,28,29]. One *AhR* gene and five *CYP1* family genes were identified in *Ciona intestinalis* type A [42]. The upstream nucleotide sequence of these *CYP1* homolog genes contains the xenobiotic responsive element that acts as the putative binding site of Ci-AhR, suggesting that Ci-AhR might transactivate *CYP1* homolog genes [42]. However, whether Ci-AhR activates transcription of *Ci-CYP1* genes is yet to be evaluated. Moreover, the detailed expression pattern of *Ci-AhR* and *CYP1* family genes remains unclear thus far.

We performed the expression analysis of AhR in *Ciona intestinalis* type A to reveal basic information on Ci-AhR function and the relationship between PAH action and AhR expression. Maternal *AhR* mRNA expression has been observed in the fertilized egg to the late tailbud stages in early development [41]. In this stage, PAHs disturbed normal development (Figure 1). These results suggest that the defects induced by these chemicals in early development might be mediated by AhR. During the metamorphosis stage, Ci-AhR was expressed in the larval, mid body axis rotation, and early juvenile stages, implying that Ci-AhR might be associated with the tail regression defect (Figure 3). Lastly, we evaluated *Ci-AhR* transcript localization in the juvenile stage. *Ci-AhR* mRNA was expressed in the digestive organ, branchial sac papillae, and dorsal tubercle, which open to the exterior of the body (Figure 4). A similar expression pattern was observed in other marine invertebrates as well. For instance, in pearl oyster *Pinctada martensii,* the *AhR* gene transcript ortholog was detected in the mantle, gill, hepatopancreas, and adductor muscle [43]. It is hypothesized that Ci-AhR is involved in metabolism of PAHs entering from seawater because these organs directly connect to the body exterior. *Ci-AhR* mRNA localization in the ganglion, also known as the ascidian central nerve, is expected to correspond to neural development and neural function observed in *Caenorhabditis elegans* [44,45,46]. Qin et al. reported that the *C. elegans* AhR homolog is involved in the regulation of aggregation behavior [46]. Collectively, the results revealed the expression pattern of *Ci-AhR* mRNA in the developmental and juvenile stages. These expression analyses were performed under normal conditions. We intend to further analyze PAH-induced changes in the level and localization of *Ci-AhR* mRNA by using quantification PCR and WISH in the next study, respectively. Such information will help to further understand the role of Ci-AhR in PAH metabolism.

## 5. Conclusions

Our study revealed that fluorene and phenanthrene affect early development and metamorphosis in *Ciona intestinalis* type A. We identified dibenzothiophene as a toxic agent in *Ciona* development as well. To the best of our knowledge, this is first report of its kind on *Ciona intestinalis* type A. Moreover, we denoted the expression of Ci-AhR during metamorphosis; in particular, tissue distribution of Ci-AhR was denoted in juvenile *Ciona*. These findings provide basic information on the molecular mechanism underlying PAH action and the biological function of AhR in marine invertebrates.

## Figures and Tables

**Figure 1 ijerph-17-01340-f001:**
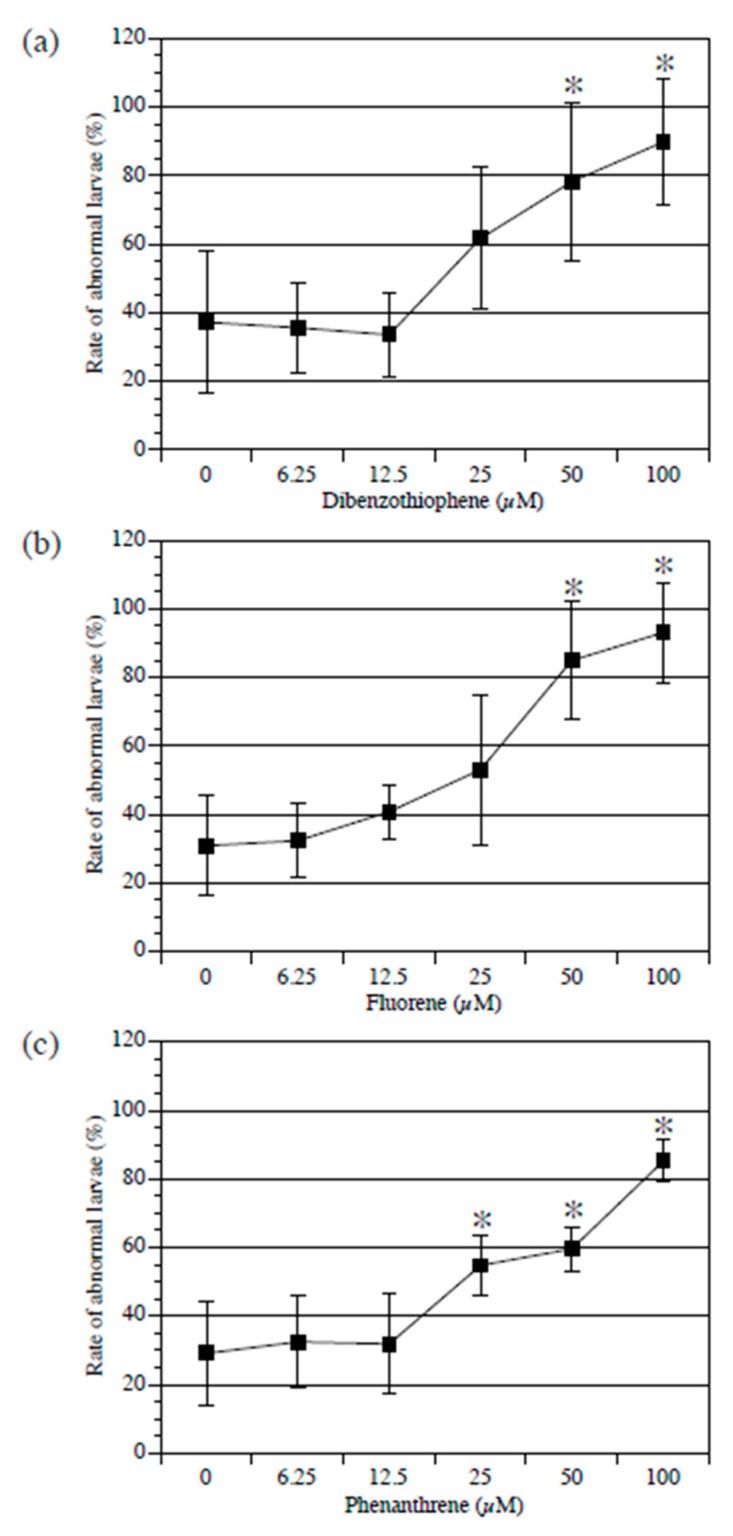
Effect of polycyclic aromatic hydrocarbons (PAHs) on *Ciona* embryogenesis. Percentage of abnormal larvae treated with different concentrations of dibenzothiophene (**a**), fluorene (**b**), and phenanthrene (**c**) from fertilized egg to swimming larva stages. Mean values of five biological replicates are presented with standard deviations. Approximately 50 embryos were examined in each experiment. Asterisks indicate significant differences (*p* < 0.05; Dunnett’s test).

**Figure 2 ijerph-17-01340-f002:**
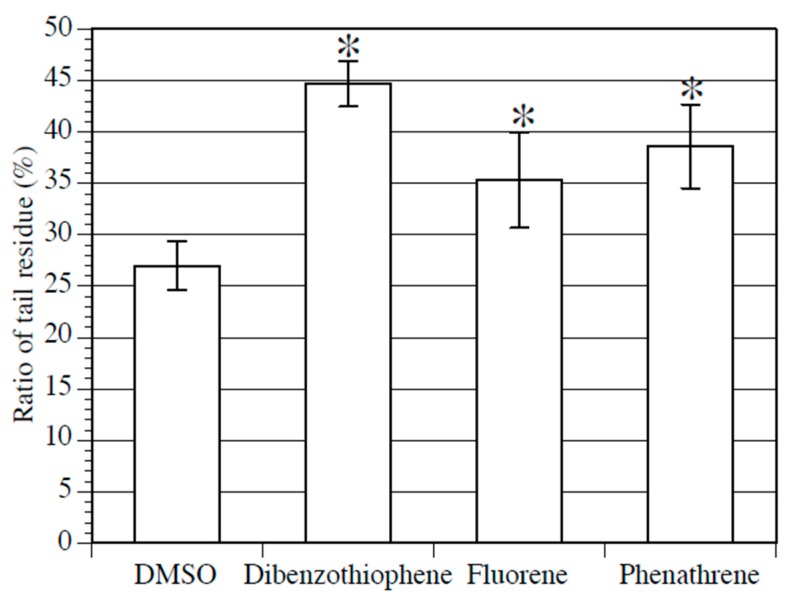
Effect of PAHs on *Ciona* metamorphosis. The ratio of tail residue diameter to body length in juveniles treated with 0.2% DMSO (negative control), 100 µM dibenzothiophene, 100 µM fluorene, or 100 µM phenanthrene from swimming larvae (20 h after fertilization) to juveniles (90 h after fertilization) stages. Mean values of five biological replicates are represented with standard deviations. Approximately 50 embryos were examined in each experiment. Asterisks indicate significant differences (*p* < 0.05; Dunnett’s test).

**Figure 3 ijerph-17-01340-f003:**
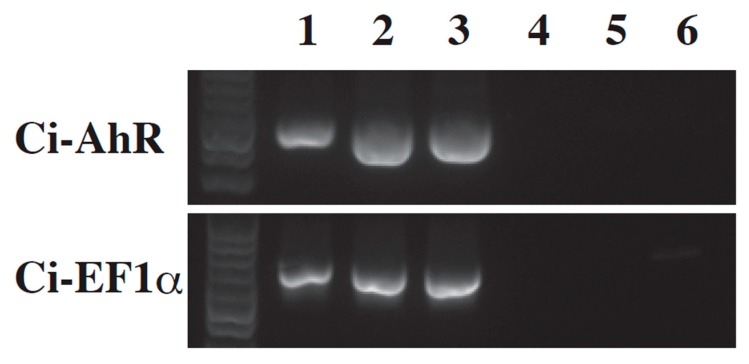
RT-PCR analysis of *Ci-AhR* and *Ci-EF1α*. *Ci-AhR* mRNA expression was determined in the swimming larva (lane 1), mid body axis rotation (lane 2), and early juveniles (lane 3) stages. No amplification was detected in the no reverse transcription template (lane 4–6). *Ci-EF1α* was amplified as an internal control.

**Figure 4 ijerph-17-01340-f004:**
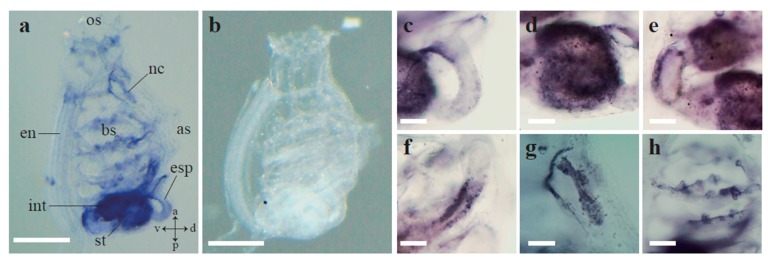
Whole-mount *in situ* hybridization (WISH) analysis of *Ci-AhR* mRNA in *Ciona* juveniles. (**a**) *Ci-AhR* mRNA localization using antisense *Ci-AhR* probe. (**b**) WISH analysis using sense probe of *Ci-AhR*. No signal was detected. Magnifications of (**a**) are presented in (**c**–**h**). a, anterior; as, atrial siphon; bs, branchial sac; d, dorsal; en, endostyle; esp, esophagus; int, intestine; nc, neural complex; os, oral siphon; p, posterior; st, stomach; v, ventral. Scale bars in (**a**,**b**) and (**c**–**h**) are 500 and 100 μm, respectively.

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
