# Peer review of "Effect of Polycyclic Aromatic Hydrocarbons on Development of the Ascidian Ciona intestinalis Type A"

_ijerph, 2020, doi:10.3390/ijerph17041340_

Round 1
Reviewer 1 Report
This is interesting research, containing valuable outcomes. Manuscript however requires proofreading since there are multiple grammatical and spelling errors. E.g. line 38: there is no such word as " air bourn" please change it to airborne;line 41- please re-write this sentence since there is grammaticaly incorect to say that " PAHs provide the variety of influence.. etc". For instance in abstract line 18, first sentence also requires change in order to be correct.
Introduction section is very short and not informative enough. It need to be extended by real up to date comparison with other researchers and outcomes of other similar studies (not only performed by Authors).
References should also be more up to date, because now it contain prevailing amount of publication older than 7 years.
Reviewer 2 Report
The influence of PAHs on marine invertebrates is a world problem. To now, there are too many published reports and papers related to this field. So the topic of this paper is very interesting. But in my mind, the background is poor. It is better to summary the related work in detail. The Ci-AhR part is too simple within qualitative analysis. And the relationship between Ci-AhR mRNA and PAHs was unclear without quantitative analysis.
On the other hand, the discussion of the influence of PAHs on marine invertebrates is also poor. The references summay is necessary and for the author's idea supporting. Finally, the conclusion is also simple and poor.
So, my suggestion is reject.
Reviewer 3 Report
Sekiguchi et al.
Influence of Polycyclic Aromatic Hydrocarbons on the Development of Ascidian, Ciona Intestinalis Type A
The authors present an analysis of the phenotypic effects in early development observed after exposure to different PHAs. The data suggest that these organisms are affected. In addition, they evaluate the expression of the Ci-AhR RNA as a potential receptor of exposure.
The following modifications are recommended to improve the manuscript:
Quality of figures needs to improve. In Figures 1 and 2, details of the experimental design such as number of biological repeats should be included.It is unclear how the Ci-EF1alpha was used as an internal control in Figure 3. Also are these non-exposed organisms? Please clarify. Same for Figure 4.
The claim that Ci-AhR acts as a sensor of pollutants is correlative. The authors should tone down these claims until betterdata are provided.
Minor changes to improve grammar:
Line 29: polluntant entry…
Line 38- air borne
Line 43 PAHs also show diverse influences including…
Line 45 : PAH, suppresses the larval…
Line 60: the generation of transgenic and genome editing technologies in ascidian organisms is also available.
Line 76: eliminate RT-PCR and add it to the end of the sentence “ were collected and dissolved in Trizol reagent for RT-PCR analysis…”
Line 155: was bent.
Line 172: was performed on swimming…
Round 2
Reviewer 2 Report
Within the author's correction, the results are much clear than before. I think it could be published in this journal after minor correction. But the question of quantitative analysis of Ci-AhR is still there. Anyway, it is a ubclear and hard problem, maybe the result should be reported in the coming future.
So, my suggestion changed to minor correction.
